# HIGH-FIDELITY AND REALTIME 3D GAUSSIAN HEAD AVATARS WITH EXPRESSIVE AND COMPACT BLENSHAPE REPRESENTATIONS

## ABSTRACT

Recent studies have combined 3D Gaussian and 3D Morphable Models (3DMM) to achieve real-time, high-quality rendering of controllable head avatars. Several techniques have attempted to express dynamic textures in facial animation when modeling 3D avatars. However, accurately capturing and displaying expressive appearance dynamics while maintaining temporal and spatial efficiency remains a technical challenge. To this end, we propose a novel method for 3D facial avatar modeling that utilizes an expressive and compact model representation, capturing dynamic facial information accurately while ensuring efficiency. We encode texture-related attributes of the 3D Gaussians in the tensorial feature representation. Specifically, we store color information of the neutral expression in static tri-planes; and represent dynamic texture details for different expressions using lightweight 1D feature lines, which are then decoded into opacity changes relative to the neutral face. Experiments show that this design introduces nonlinear expressiveness to the model, enhancing its performance, while the compact representation maintains real-time rendering capabilities and significantly reduces storage costs. This approach thus broadens the applicability to more scenarios.

## 1 INTRODUCTION

Photorealistic 3D avatars generation is a fundamental research topic within the field of computer graphics and computer vision, encompassing a variety of applications including movies, gaming, and AR/VR etc. A high-quality 3D human head avatar needs to meet several requirements: 1) achieve photorealistic high-fidelity rendering from arbitrary viewpoints; 2) generate animations of various expressions conveniently; and 3) ensure real-time rendering with reduced memory and storage. However, achieving the above quality and performance poses significant technical challenges.

Some early methods are based on 3D face Morphable Models (3DMM) Blanz & Vetter (1999); Thies et al. (2016) to achieve 3D avatar generation, which can easily control facial expressions using low-dimensional PCA coefficients, but the rendering effect lacks realism. With the development of NeRF Mildenhall et al. (2021), some approaches combine NeRF and 3DMM to achieve high-quality, animatable head avatars, but the volumetric rendering process of NeRF requires substantial computation, making it difficult to achieve real-time performance. Recently, 3D Gaussian Splatting (3DGS) Kerbl et al. (2023) has gained widespread attention for its high-quality rendering and real-time performance, with some works attempting to combine 3DGS with 3DMM to achieve real-time, highly realistic, and animatable head avatars. It is important to model the *dynamic facial textures and details* that change with expressions, but representing such details inevitably introduces additional time and memory costs. Some methods Ma et al. (2024) explicitly store a set of Gaussian splats for each expression, resulting in significant storage requirements. Alternatively, some methods Xu et al. (2024) employ MLPs to implicitly model dynamic textures, followed by super-resolution techniques to enhance detail rendering, but at the cost of not being able to render in real-time.

To address these issues, we propose a high-fidelity 3D head avatar reconstruction method that takes multi-view face videos as inputs and outputs an animatable 3D head avatar with dynamic facial textures and details, while maintaining space efficiency and real-time speed.

Our method uses 3DGS to model the dynamic head avatars, and combines three different representations, *i.e.,* parametric mesh, triplane, and lightweight feature lines, to describe various aspects of 3D head information. Firstly, the geometric attributes of Gaussian splats and large-scale head movements are **bound with a parametric mesh model** (FLAME tracked from the multi-view face video). Next, observing that neighboring Gaussian splats share similar appearances and dynamics, we store texture-related information, including both canonical and dynamic textures, in a tensorial feature representation, reducing spatial redundancy and optimizing storage. The static textures of the neutral expression are stored in a **triplane** within the canonical space, replacing the spherical harmonic coefficients used in the 3DGS. For the dynamic textures, for each expression blendshape, we store a neural grid representing the opacity offsets relative to the neutral face.

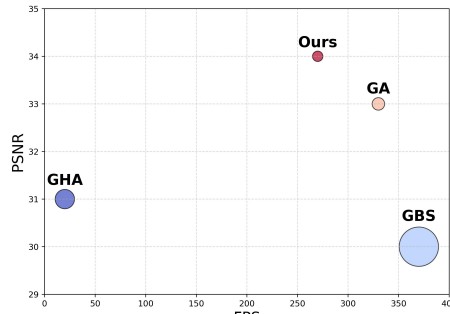

Figure 1: Our method improves PSNR for novel view synthesis tasks while ensuring real-time performance and low storage. The area of the points in the figure is proportional to the square root of the storage space.

Given the large number of blendshapes, storing 3D tensors or 2D planes incurs excessive storage costs. To enable a compact representation for dynamic textures, we use a more **lightweight 1D feature line** as an alternative. The feature lines are linearly interpolated using blendshape coefficients, and a non-linear MLP decoder outputs the opacity offset. Finally, each Gaussian splat's geometry attributes along with canonical and dynamic textures, form the complete 3D Gaussian attributes used for image rendering. Additionally, to address the issue of unbalanced training data (where large-expression frames are scarce), we compute the similarity between frames based on pre-computed FLAME meshes, cluster frames based on the similarity matrix, and uniformly sample within each cluster during training, thereby improving the reconstruction quality of large expressions.

We conduct experiments on the Nersemble Kirschstein et al. (2023) dataset, which demonstrates that our method accurately reconstructs dynamic facial texture details and improves the quantitative metrics of the rendered images. Our compact representations require no more than 10MB per subject, making it the most storage-efficient method compared to the state-of-the-art competitors. Additionally, we achieve 220 FPS, ensuring real-time performance. The spatial and temporal efficiency of our approach allows it to be extended to broader application scenarios, such as fast network transmission and real-time rendering in mobile video conferencing.

## 2 RELATED WORKS

### 2.1 DYNAMIC SCENE REPRESENTATION

NeRF Mildenhall et al. (2021) and 3D Gaussian Splatting (3DGS) Kerbl et al. (2023) are widely used for novel view synthesis and multi-view reconstruction for static scenes. To adapt these methods for dynamic scenes, most approaches reconstruct a scene at a specific moment as the canonical frame, and use additional representations to capture dynamic changes. Some methods Park et al. (2021b;a); Wu et al. (2024) use MLPs to model displacement relative to the canonical frame. Other methods use spatially or temporally discrete representations. To reduce the storage demands of high-dimensional data, compact formats like K-Plane Fridovich-Keil et al. (2023) or Tensor4D Shao et al. (2023) compress 4D space into multiple 2D planes. Low-dimensional parametric models, such as sparse control points Huang et al. (2024); Jiang et al. (2024) and parametric curves Wang et al. (2021); Li et al. (2024), are also used to enhance training stability. In the fields of 3D human body and face reconstruction, some mature shape prior models for the human body Loper et al. (2023) and face Li et al. (2017); Paysan et al. (2009) use low-dimensional parameters to accurately describe geometry and motion, making them easy to control. Many methods combine these priors with NeRF and 3DGS for animatable human or head avatars. However, the low-dimensional parameters limit their expressiveness, and some new representations Li et al. (2023) are proposed to capture finer details. We will discuss head avatars reconstruction in more details in the next section.

## 2.2 ANIMATABLE HEAD AVATAR

Some 3D animatable head avatar reconstruction methods Zielonka et al. (2023); Gafni et al. (2021); Xu et al. (2023) use NeRF to implicitly represent geometry and take tracked 3DMM coefficients as inputs. In contrast, 3DGS uses explicit geometric primitives that can be directly bound to a mesh. GA Qian et al. (2024) defines Gaussian splats in the local coordinate system of the mesh triangles, allowing splats to move with the mesh. However, since GA uses the same set of Gaussian splats for all expressions, it fails to accurately model dynamic details that change with expressions. 3DGS-Blendshapes Ma et al. (2024) optimizes a set of 3D Gaussians splats for each blendshape, enabling linear interpolation of Gaussian attributes using blendshape coefficients, but this demands significant storage. Moreover, linear interpolation struggles to capture complex dynamic details. GHA Xu et al. (2024) uses a coarse mesh and two MLPs to predict geometry and color offsets, and converts 3D Gaussian features into RGB images via a super-resolution module. NPGA Giebenhain et al. (2024b) leverages the rich latent expression space and detailed motion prior of MonoNPHM Giebenhain et al. (2024a) and employs a screen-space CNN to suppress small-scale artifacts. GHA Xu et al. (2024) and NPGA Giebenhain et al. (2024a) achieve high-quality detailed rendering, but fail to achieve real-time rendering speed. In contrast, we achieve high-quality rendering with dynamic details, real-time speed, and compact storage simultaneously, by combining different representations including mesh, triplanes and feature lines to capture different aspects of 3D animatable avatar.

## 2.3 COMPACT 3D SCENE REPRESENTATIONS FOR NOVEL VIEW SYNTHESIS.

NeRF uses deep MLPs to model scenes, but suffers from slow inference and scalability issues for large, unbounded areas. Some methods Müller et al. (2022); Chan et al. (2022) split the large MLP into spatially discrete features and tiny MLP decoders, improving inference speed and training efficiency. Other approaches decompose large NeRFs into smaller ones Yang et al. (2021) to reduce storage needs. 3DGS is efficient in rendering but requires storing a large number of Gaussian splats with their attributes for accurate scene representation. Recent approaches address this issue by using region-based vector quantization Niedermayr et al. (2024), K-means-based codebooks Navaneet et al. (2023), or learned binary masks for each Gaussian. They also replace spherical harmonics (SHs) with grid-based neural networks Lee et al. (2024); Zou et al. (2024) within an adjusted training framework to minimize model size. In this paper, we leverage a lightweight feature line for each expression blendshape, to enable a compact representation for dynamic facial textures.

## 3 METHOD

Our method takes multi-view face videos as inputs and outputs an animatable head avatar with dynamic textures. As shown in Fig. 2, our method uses 3DGS to model the dynamic head avatars, and stores large-scale geometry movements of head, head appearance in the canonical space, and dynamic texture variations for each blendshape using three different structures, *i.e.,* mesh, triplane, and feature lines, respectively. 1) Geometry movements bound to mesh: For each frame in the video, we track a FLAME Li et al. (2017) mesh from multi-view observations and known camera parameters by a photometric head tracker Qian et al. (2024). We follow Qian et al. (2024) to generate geometry attributes (including position, rotation and scaling) of splats via deformed mesh for each frame. 2) Appearance in canonical space stored in a Triplane: Instead of using **SH** to represent view-dependent color as in original 3DGS, a triplane is used to store radiance around the 3D head in the canonical space. The features sampled from the triplane, along with the view direction transformed into the canonical space, are fed into a tiny MLP decoder to obtain the color of the Gaussian. 3) Dynamic textures stored in Feature Lines: We utilize a feature line per expression blendshape to store dynamic textures. The tracked expression coefficients of each frame are used to interpolate the feature lines. An MLP decoder then maps the features sampled from the interpolated feature line into an opacity offset, which is added to canonical opacity. Finally, the aforementioned gaussian attributes are combined to render the image, which is compared with the ground truth.

### 3.1 PRELIMINARIES

**3DGS.** 3D Gaussian Splatting Kerbl et al. (2023) enables novel view synthesis of a static scene using anisotropic 3D Gaussians based on multi-view images and camera parameters. A scene is

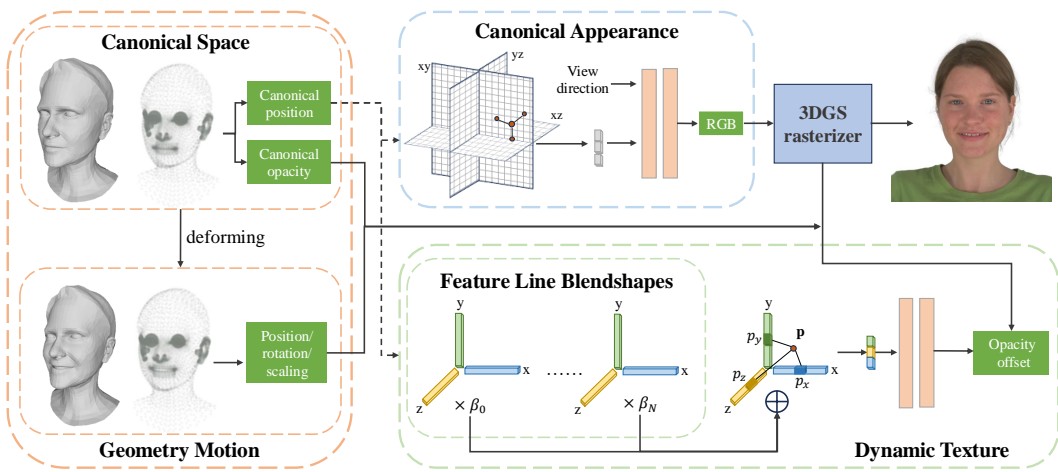

Figure 2: Our goal is to reconstruct 3D Gaussian head avatar with dynamic textures, ensuring real-time performance and minimized storage. We use a parametric face model to describe large-scale geometry deformations, moving the bound Gaussian splats accordingly. A triplane stores view-dependent head appearance in canonical space, while compact feature lines are used for dynamic textures per blendshape, allowing interpolation with expression coefficients. Finally, the geometry attributes of the Gaussian, along with the canonical appearance and dynamic textures, are combined to render the face image.

represented by a collection of 3D Gaussian splats, each defined by a covariance matrix $\Sigma$ centered at a 3D point $\mu$. The covariance matrix $\Sigma$ is further decomposed into a scaling matrix $S$ and a rotation matrix $R$ to ensure $\Sigma$ to be semi-definite through differentiable optimization, which is formulated as:

$$G(\mathbf{x}) = e^{-\frac{1}{2}(\mathbf{x}-\mu)^T \Sigma^{-1}(\mathbf{x}-\mu)} \qquad \Sigma = RSS^T R^T. \tag{1}$$

Each 3D Gaussian contains the following attributes: position vector $\mu \in \mathbb{R}^3$, scaling vector $\mathbf{s} \in \mathbb{R}^3$, quaternion $\mathbf{q} \in \mathbb{R}^4$, opacity value $\alpha \in \mathbb{R}$ and spherical harmonics $\mathbf{SH} \in \mathbb{R}^{(k+1)^2 \times 3}$ to represent view-dependent color (where $k$ denotes the degree of freedom). In our paper, we use $\mathbf{r} \in \mathbb{R}^{3\times3}$ to notate the corresponding rotation matrix to $\mathbf{q}$. The final color for a given pixel is calculated by sorting and blending the overlapped Gaussians:

$$\mathbf{C} = \sum_{i=1}^{N} c_i \alpha_i' \prod_{j=1}^{i-1}(1 - \alpha_j'), \tag{2}$$

where blending weight $\alpha'$ is given by evaluating the 2D projection of the 3D Gaussian multiplied by a per-point opacity $\alpha$.

**Gaussian Avatars.** Qian et al. (2024) binds Gaussian splats to a tracked mesh, which is obtained by fitting FLAME (a facial prior model) parameters to multi-view observations. Each 3D Gaussian splat is paired with a mesh triangle. *I.e.,* the geometric attributes of a splat, including the position $\mu$, rotation $\mathbf{r}$, and anisotropic scaling $\mathbf{s}$, are defined in the local space of the corresponding mesh triangle. These locally defined geometric attributes are optimized during training, and once optimized, the relative position between a splat and the corresponding mesh triangle is unchanged, which makes the splat static in the triangle's local space, but dynamic in the global space as the triangle moves. Given the three vertices of a triangle, the average vertex position $T$ is set as the local origin. A rotation matrix $R$ is constructed by concatenating the direction of one edge, the normal vector of the triangle, and their cross product as column vectors, to represent the orientation change from local space to global space. A scalar $k$ is computed as the average length of one of the edges and its perpendicular, to represent the triangle's scaling. Then the transformation from the local space to the global space is conducted as:

$$r' = Rr, \qquad \mu' = kR\mu + T, \qquad s' = ks. \tag{3}$$

## 3.2 Appearance in Canonical Space via Triplane

In 3DGS, 48 out of the total 59 parameters for each Gaussian are used for spherical harmonics (SH, up to 3 degrees) to capture view-dependent color. Based on the observation that neighboring splats have similar appearance, instead of storing 48 SH parameters for each splat, we use a triplane to store implicit encodings, and a tiny MLP decoder to decode the encodings to RGB colors, which compresses the model size and accelerates the inference speed.

The triplane $T$ consists of three orthogonal feature planes aligned with the axes: $\{T_{xy}, T_{xz}, T_{yz}\} \in \mathbb{R}^{3 \times n_f \times n_f \times n_{d1}}$, where $n_f \times n_f$ is the spatial resolution of the 2D feature plane, and $n_{d1}$ is the feature dimension. For any given position $\mathbf{p}$ in canonical space, the corresponding feature vector is obtained by projecting the position onto the axis-aligned planes (the $x - y$, $x - z$ and $y - z$ planes), interpolating to obtain the features on each feature plane and concatenating the interpolated features, which is formulated as:

$$t(\mathbf{p}) = interp(T_{xy}, \mathbf{p}_{xy}) \oplus interp(T_{xz}, \mathbf{p}_{xz}) \oplus interp(T_{yz}, \mathbf{p}_{yz}),$$

where *interp* represents trilinear interpolation, $\oplus$ represents concatenation, and $\mathbf{p}_{xy}, \mathbf{p}_{xz}, \mathbf{p}_{yz}$ refer to the projected position on each plane.

Note that the triplane is defined in the *canonical space*, which means the global space corresponding to the neutral expression. Meanwhile, we call the global space corresponding to the non-neutral expression in each frame as the *deformed space*. Given a splat with the position $\mu$, rotation $\mathbf{r}$, and scale $\mathbf{s}$ of defined in the local space, it can be transformed into the canonical space and deformed space of the current frame based on the canonical transformation $(R_c, T_c, k_c)$ and the deformed transformation $(R_d, T_d, k_d)$ respectively. The local coordinate system can also serve as a bridge to transform between the deformed space and the canonical space. The transformation from view direction in the deformed space (denoted as $\mathbf{v}_d$) to view direction in the canonical space (denoted as $\mathbf{v}_c$) can be formulated as:

$$\mathbf{v}_c = R_c R_d^{-1} \mathbf{v}_d.$$

Finally a tiny MLP decodes $t(\mathbf{p})$ and $\mathbf{v}_c$ into RGB value $\mathbf{c}$, which we use as the first three components of degree-1 spherical harmonics for 3DGS rendering. Additionally, since the majority of facial information is concentrated in the frontal view, with less information available from the side views, we utilize a larger feature dimension in the $T_{xy}$ while employing lower dimension for $T_{xz}, T_{yz}$ to achieve storage compression.

## 3.3 Dynamic Texture via Feature Line Blendshapes

Some existing researches have demonstrated that the color attributes of 3D Gaussians can be compressed using triplanes, proving that the colors of Gaussian points are consistent within a small neighborhood. We aim to extend this conclusion of local consistency from static scenes to dynamic scenes, achieving efficient representation of dynamic textures. For each expression blendshape, we store a separate representation that describes the texture changes (opacity offsets) relative to the neutral expression induced by that expression. Since the FLAME model encompasses 100 PCA blendshapes, storing a triplane or tensor for each blendshape incurs excessive memory consumption. To enable a compact representation for dynamic textures, for each blendshape $i$, we use a lightweight 1D feature line to encode the texture changes of that expression, denoted as $(L_x^i, L_y^i, L_z^i) \in \mathbb{R}^{3 \times n_{d2} \times n_s}$, where $n_s$ represents the length of the 1D feature line, and $n_{d2}$ represents the feature dimension. For frame $j$, we calculate the feature line of the expression in this frame by interpolating $n_b$ feature lines with the blendshape coefficients $\beta_j \in \mathbb{R}^{n_b}$ where $n_b$ is the number of blendshapes, which is formulated as:

$$l_b^j = \sum_{i=0}^{n_b} \beta_j^i * (L_x^i, L_y^i, L_z^i).$$

In addition to the linear expression basis, FLAME incorporates a nonlinear quaternion jaw rotation to describe large-scale jaw movements. To unify the linear basis with the nonlinear rotation, we follow the method proposed in Li et al. (2023), extracting linear jaw rotation bases $\{\mathbf{q}_k : k \in \{0, \ldots, n_j\}\}$ from all video frames via farthest point sampling. And we store a feature line $(L_x^k, L_y^k, L_z^k)$ for each

jaw rotation basis. Then the distance between the jaw rotation of frame $j$ and the $k$-th jaw rotation basis is calculated as $d(j, k) = 1 - |\mathbf{q}_j^T \mathbf{q}_k|$, where both $\mathbf{q}_j$ and $\mathbf{q_k}$ are unit quaternion. And the jaw feature lines are interpolated using inverse distance weighting to calculate the feature line of the jaw rotation, formulated as:

$$l_r^j = \sum_{k=0}^{n_j} \beta_j^k * (L_x^k, L_y^k, L_z^k), \qquad \beta_j^k = \frac{1 - d(j, k)}{\sum_{k=0}^{n_j}(1 - d(j, k))}.$$

Similar to the triplane, the feature vector of a given position $\mathbf{p}$ in frame $j$ is calculated by projecting the position onto the $x, y$ and $z$ axes, interpolating to obtain the features on each feature lines, and concatenating the interpolated features, formulated as:

$$l^j(\mathbf{p}) = interp(l_x^j, \mathbf{p}_x) \oplus interp(l_y^j, \mathbf{p}_y) \oplus interp(l_z^j, \mathbf{p}_z).$$

This projection and interpolation process is applied to both the expression blendshape feature line $l_b^j$ and the jaw rotation feature line $l_r^j$.

We finally utilize a tiny MLP $\theta$ to decode interpolated $l_b^j(\mathbf{p})$ and $l_r^j(\mathbf{p})$ into opacity offset $\Delta\alpha$, which will be added to the canonical opacity $\alpha_c$ (opacity of the neutral experssion). The final opacity is calculated as:

$$\alpha = \alpha_c + \Delta\alpha = \alpha_c + \theta(l_b^j(\mathbf{p}), l_r^j(\mathbf{p})).$$

In the PCA expression basis of FLAME model, the facial motion caused by facial expressions are primarily concentrated in the leading components. Our experiments show that using only the leading expression coefficients achieves similar results, allowing us to reduce the number of feature lines, thereby reducing storage.

### 3.4 TRAINING

**Class-balanced Sampling.** In the training dataset, most images exhibit small facial movements, with only a small portion showing significant movements, which makes the reconstruction of large expressions less accurate. However, the distribution of the training data is unknown, and simply increasing the sampling probability for images with large expressions may introduce bias against images with smaller expressions. Instead, we propose a class-balanced sampling method based on pre-computed FLAME parameters to prioritize frames with larger expressions during training.

First, we measure the similarity between two frames by comparing the differences in vertex displacements of the FLAME mesh. Given that different regions of the face exhibit varying degrees of motion, *e.g.,* the lips moving significantly more than the eyes, we empirically increase the weight of the vertices around the eyes. We denote the FLAME mesh for frame $i$ as $M_i$, and the similarity score between frame $i$ and frame $j$ is calculated as $dist(i, j) = ||M_i - M_j||_2^2 * \mathbf{w}$, where $\mathbf{w}$ denotes the weight of each vertex. Next, we perform spectral clustering on the similarity matrix to categorize all frames into $n$ classes. Finally, we randomly sample from each category, ensuring a uniform sampling probability distribution between categories and equal probability for samples within each category.

**Loss Function.** We use the L1 loss and D-SSIM loss between the rendered images and the ground truth images as image supervision, which can be formualted as

$$\mathcal{L}_{image} = (1 - \lambda)\mathcal{L}_1 + \lambda\mathcal{L}_{\text{D-SSIM}}.$$

Assuming that the Gaussian splats should roughly conform to the mesh and be similar in size to the bound triangles, we follow methods proposed by Qian et al. (2024) to employ a position loss and a scale loss to prevent splats from being excessively far from the mesh or excessively large.

$$\mathcal{L}_{geom} = \lambda_{position}\mathcal{L}_{position} + \lambda_{scaling}\mathcal{L}_{scaling}. \tag{4}$$

As the scalp and teeth having low correlation with the expression coefficients, we add a normalization term to constrain opacity offset of splats bound to scalp and teeth triangles to be zero.

$$\mathcal{L}_{norm} = \lambda_{hair}|\Delta\alpha_{hair}| + \lambda_{teeth}|\Delta\alpha_{teeth}|.$$

The training loss can be formulated as the following equation:

$$\mathcal{L} = \mathcal{L}_{image} + \mathcal{L}_{geom} + \mathcal{L}_{norm},$$

where $\lambda = 0.2, \lambda_{position} = 0.01, \lambda_{scaling} = 1, \lambda_{hair} = 1, \lambda_{teeth} = 1, .$

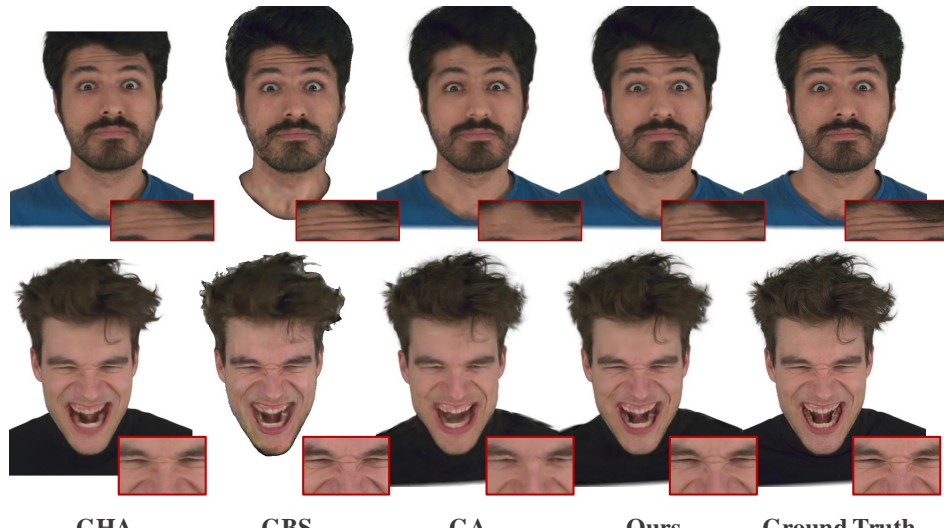

|   GHA   |   GBS   |   GA   |   Ours   |   Ground Truth   |

Figure 3: Qualitative comparison with baseline methods on novel view synthesis task.

# 4 EXPERIMENTS

## 4.1 SETTINGS AND DATASET

We conduct experiments on nine individuals from the nersemble dataset, collecting a total of 11 video segments of each subject from 16 different viewpoints. Each participant performs 10 distinct expressions and emotions as instructed, followed by a free performance in the last video segment. The videos are downsampled to a resolution of $802 \times 550$. We utilize the FLAME coefficients and camera parameters provided in GAQian et al. (2024), including shape $\beta$, translation $\mathbf{t}$, pose $\theta$, expression $\psi$, and vertex offset $\Delta \mathbf{v}$ in the canonical space.

We compare the experimental results across three tasks: 1) Novel View Synthesis: 15 out of 16 viewpoints are used for training, while the remaining viewpoint is reserved for testing; 2) Self-Reenactment: Testing is conducted using videos of the same individual showcasing unseen poses and expressions from all 16 viewpoints; 3) Cross-Identity Reenactment: An avatar is driven by the motions and expressions of other individuals. We randomly select one video from 10 prescribed videos for testing of task 2 and test on task 3 using free-performance sequences.

## 4.2 IMPLEMENTATION DETAILS

We implemented the code using PyTorch and trained for 600,000 iterations using the Adam optimizer for each subject. Both the triplane and feature lines consist of two components: the neural grid feature and the MLP decoder. We trained triplane and feature line blendshapes with the same learning rates, setting learning rate of feature to $1e-3$ and the MLP learning rate to $1e-4$. Each plane of the triplane is $64 \times 64$ in size, with the feature dimension of the $T_{xy}$ being 64, and the feature dimensions of the $T_{xz}, T_{yz}$ planes set to 32. The MLP decoder following triplane consists two-layer with 128 dimensions per layer, using ReLU as the activation function. Additionally, we apply position encoding to improve the resolution of view direction. We assign feature lines with a spatial resolution of 64 and feature dimension of 32 to the first 80 PCA expression bases and 16 key jaw rotation bases. The decoder for the feature lines is a two-layer MLP with 128 dimensions per layer. The triplane requires 2.08M of storage, the feature line requires 2.41M, and the other Gaussian attributes (including position/rotation/scaling and canonical opacity) average 5.2M per subject.

## 4.3 COMPARISON

We compare the performance of three baselines on 3D Gaussian splatting for head avatar reconstruction, including GAQian et al. (2024), GHAXu et al. (2024), GBSMa et al. (2024). GBS Ma et al. (2024) use monocular face videos as input, reconstructing only the head while not reconstructing the clothing and shoulders. To ensure a fair comparison, we use multi-view videos and the FLAME

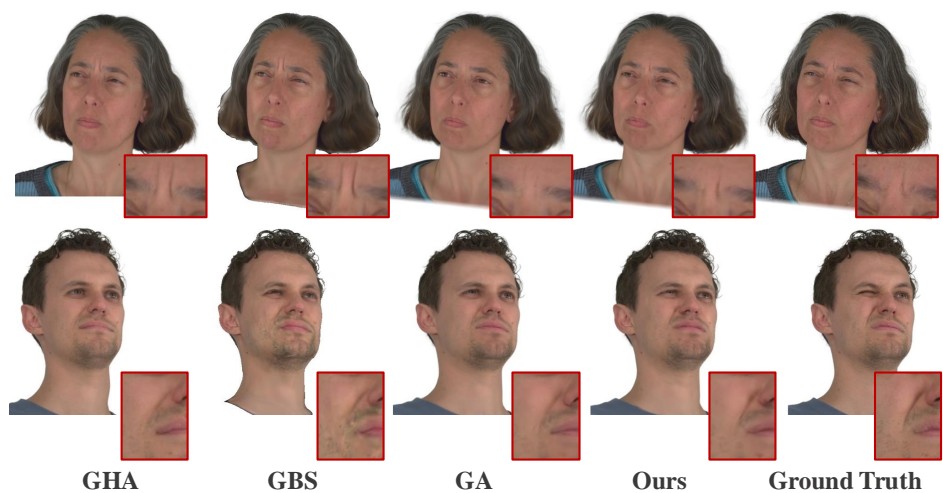

| GHA | GBS | GA | Ours | Ground Truth |

Figure 4: Qualitative comparison with baseline methods on self-reenactment task.

parameters tracked from the multi-view videos following GA Qian et al. (2024) to serve as input of GBS, and segment the clothing area using face parsing, excluding it from the quantitative metrics.

We employ PSNR, SSIM, and LPIPS Zhang et al. (2018) as quantitative metrics while listing the required storage size and FPS, which can be found in 1. The storage size does not include pre-tracked 3DMM parameters. The FPS is tested on NVIDIA RTX4090 GPU. Our method occupies the minimum storage, ensuring real-time rendering speed, while also improving PSNR metrics on the tasks of novel view synthesis and self-reenactment.

| | Novel View Synthesis | | | Self-Reenactment | | | Performance | |
|---|---|---|---|---|---|---|---|---|
| Method | PSNR↑ | SSIM↑ | LPIPS↓ | PSNR↑ | SSIM↑ | LPIPS↓ | Storage | FPS |
| GA | 33.1416 | 0.9538 | 0.04492 | 30.4404 | 0.9409 | 0.04969 | 21M | 330 |
| GHA | 31.0704 | 0.9665 | 0.03653 | 30.0030 | 0.9391 | 0.03975 | 120M | 20 |
| GBS | 30.0097 | 0.9525 | 0.05853 | 28.1952 | 0.9420 | 0.06708 | 2G | 370 |
| **Ours** | 33.9700 | 0.9569 | 0.04484 | 30.5058 | 0.94010 | 0.05219 | 10M | 270 |

Table 1: Quantitative comparison with baselines. GREEN indicates the best of all methods. YELLOW indicates the second.

Fig 3 demonstrates the effectiveness of our method on novel view synthesis task. Compared to GA and GHA, our approach better reconstructs dynamic textures generated by significant expressions, such as the forehead wrinkles in the first row and the wrinkles around the right eye in the second row. GBS linearly interpolates all Gaussian attributes using expression coefficients, which can result in artifacts during large-scale nonlinear movements, such as on the left side of the chin in the second row. Our method achieves better modeling performance in non-facial regions compared to GBS, as demonstrated by the neck in the first row.

Fig 4 shows the qualitative comparison results for the self-reenactment task. Our method renders more accurate appearance from different viewpoints, while also demonstrating better generalization ability, avoiding overfitting of the wrinkles, such as the frown lines in the first row of Fig 4. Fig 5 illustrates the performance of our method in the cross-reenactment task, showcasing the generation of distinct wrinkle effects.

### 4.4 ABLATION STUDY

**Effects of tensorial texture representations.** We performed ablation studies on subject #306 to assess the impact of triplane and feature line. In Table 2, the first row reflects the use of SH coefficients per Gaussian splat. The second row uses triplane for canonical texture, excluding dynamic textures. Triplane improves self-reenactment performance by mitigating expression overfitting, though it de-

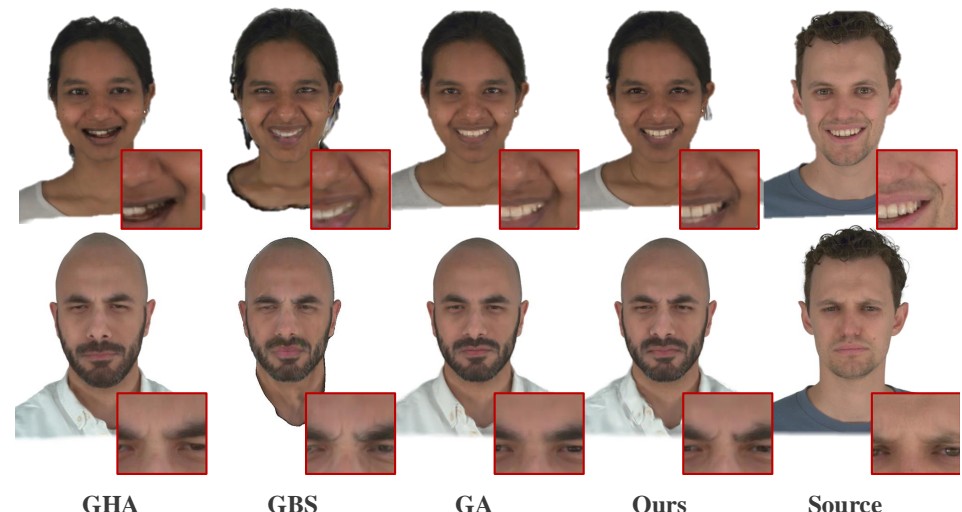

| GHA | GBS | GA | Ours | Source |

Figure 5: Cross-identity reenactment of head avatars. We use the expression and pose of the source subject on the far right to drive the character on the left.

creases novel view synthesis metrics, and triplane may blur high-frequency details, resulting in a decreased LPIPS metric.

| Method | Novel View Synthesis | | | Self-Reenactment | | |
|---|---|---|---|---|---|---|
| | PSNR↑ | SSIM↑ | LPIPS↓ | PSNR↑ | SSIM↑ | LPIPS↓ |
| w/o trip, w/o fl | 36.3942 | 0.9784 | 0.01898 | 33.4741 | 0.9672 | 0.02321 |
| w. trip, w/o fl | 35.1206 | 0.9750 | 0.02506 | 34.0785 | 0.9685 | 0.02720 |
| w/o sampling | 38.1641 | 0.9804 | 0.01846 | 34.1067 | 0.9685 | 0.02492 |
| **Ours** | 37.9416 | 0.9801 | 0.01866 | 34.2766 | 0.9693 | 0.02416 |

Table 2: Ablation Study on subject #306. "trip" refers to triplane which stores canonical appearance. "fl" refers to feature line blendshapes.

**Ablation on Class-Rebalanced Sampling.** We conduct ablation experiments on the effect of class-rebalanced sampling, as shown in Table 2. Frames with large expressions are less frequent in the training data compared to frames with small motion. The resampling method prevents the model from overfitting to frames with small motion, thereby improving generalization to unseen expressions.

**Normalization on opacity offset of hair and teeth.** Regions like hair and teeth are not strongly correlated with facial expressions. To address this, we introduce a regularization term that constrains the opacity offsets of Gaussian points bound to these areas to remain small. Fig. 6 demonstrates the effectiveness of this constraint. Without it, floaters may appear around the hair, and artifacts may occur in the teeth area during expression changes.

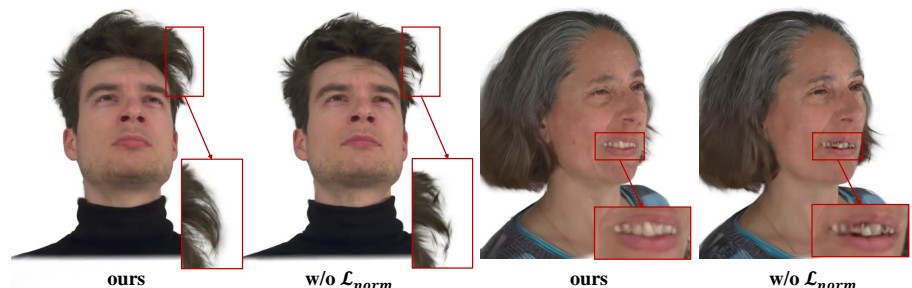

| ours | w/o $\mathcal{L}_{norm}$ | ours | w/o $\mathcal{L}_{norm}$ |

Figure 6: Opacity offset normalization of hair and teeth helps prevent artifacts when self-reenactment.

## 4.5 CONCLUSION

In this paper, we propose a 3D Gaussian Splatting head avatar modeling method that balances dynamic details capturing with real-time performance and low storage. The use of compact tensorial feature representations (static triplane for canonical appearance, and lightweight 1D feature lines for dynamic textures) allows for accurate expression modeling, leading to improved rendering quality. Experimental results demonstrate that our approach not only enhances the expressiveness of head reconstruction but also maintains real-time rendering speed and minimal storage, making it suitable for a wide range of practical applications.

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

# A SUPPLEMENTARY MATERIALS

## A.1 BASELINES

We conducted comparative experiments with three baseline methods: GA, GHA, and GSBlend-shapes. Both GHA and our method utilize multi-view videos from the Nersemble dataset, but the input resolutions differ. GHA processes 2K resolution images, while our input images are down-sampled by a factor of four. To ensure fairness in testing, we first downsampled the 2K images by four times and then upsampled them back to their original size as the input for GHA.

GSBlendshapes is a monocular facial video reconstruction method, requiring the use of a metrical tracker to regress FLAME coefficients and camera parameters from the images, which serve as the model's input. In our approach, the input consists of multi-view videos along with camera parame-ters and FLAME coefficients tracked from the multi-view data. To align the inputs, we concatenate the multi-view videos into a single video and use the multi-view tracked FLAME coefficients in place of the metrical tracker's results. It is important to note that the metrical tracker uses the 2020 version of FLAME, with two additional expression bases for describing closed eyes, while we use the 2023 version of FLAME.

To align the inputs, we use FLAME 2023 and the tracked coefficients from multi-view videos to generate meshes, then optimize the FLAME 2020 parameters by calculating the mesh vertex posi-tions loss. Note that the parameters output by the metrical tracker do not include the hair offset or neck motion, so we calculate the loss using only the facial region vertices. First, we compute the shape coefficients using the neutral expression, then iteratively regress the expression coefficients, eye rotation, jaw rotation for each frame.

## A.2 IMPLEMENTATION DETAILS.

We show the jaw rotation basis extracted from all frames in videos of id #074 in Fig 7, and we show the cluster center for expression rebalanced sampling.

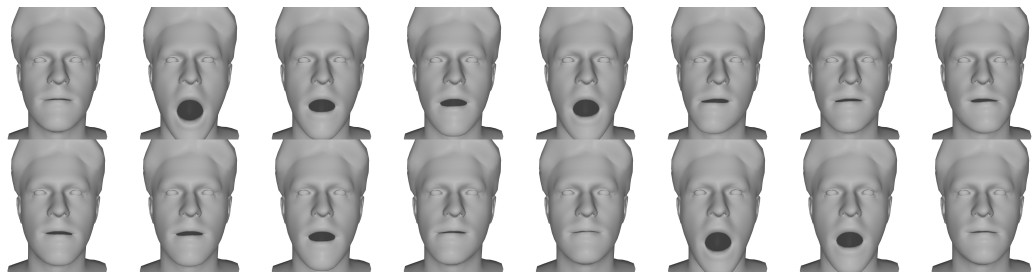

Figure 7: Basis jaw rotation extracted from all frames from videos via farthest point sampling of subject#074.

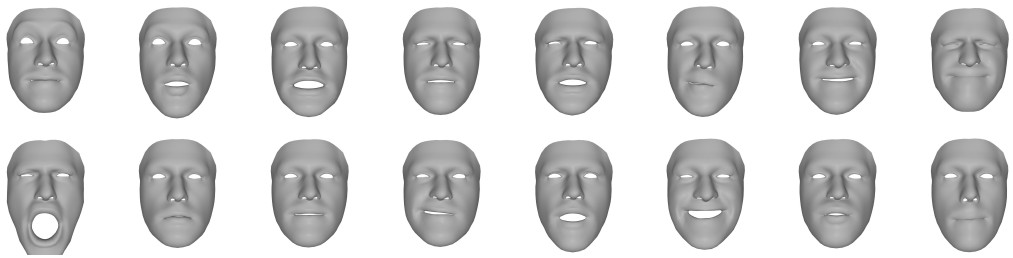

Figure 8: Cluster center for expression rebalanced sampling of subject#074.

