# OpenReview forum: "High-fidelity and realtime 3D Gaussian Head Avatars with Expressive and Compact blenshape representations"
_ICLR.cc/2025/Conference — ICLR 2025 Conference Withdrawn Submission_

### Official Review · Reviewer_N1kF · 2024-10-30

**Soundness:** 3
**Presentation:** 4
**Contribution:** 2
**Rating:** 5
**Confidence:** 4

**Summary:**

This paper addresses the issues of time and storage cost in the novel view synthesis task by proposing the use of triplanes instead of spherical harmonic coefficients in 3D Gaussian splatting to represent colors. Additionally, it employs lightweight 1D features to enable a compact representation for dynamic textures. The method binds 3D Gaussians to the FLAME mesh and ultimately uses 3DGS to render avatar images for novel view synthesis. Experiments demonstrate the effectiveness of this approach in terms of some rendering fidelity and storage efficiency.

**Strengths:**

1. The paper uses triplanes instead of traditional spherical harmonic coefficients in 3DGS to represent colors, reducing the number of parameters required for model training. This approach introduces some novelty.
2. The paper employs lightweight 1D features to enable a compact representation for dynamic textures, making the model more compact.
3. In terms of PSNR and storage, experiments demonstrate the effectiveness of the method in both rendering quality and compactness.

**Weaknesses:**

1. The paper is motivated by achieving temporal and spatial efficiency; however, experimental results show that the FPS of this method does not surpass GA and GBS.
2. In terms of visualization accuracy, as shown in Fig. 3, Fig. 4, and the demo video, the results of this method do not seem to significantly improve compared to baselines like GBS. For example, in Fig. 3, the expression of facial wrinkles is actually more detailed with GBS.
3. The method only improves the representation of RGB and opacity, offering limited contributions. Overall, it lacks novelty in technical aspects.
4. In Fig. 2, the arrow between the 3DGS rasterizer and the Opacity offset seems to be reversed.
5. The paper includes few baselines and does not compare with NeRF-based methods, such as "Neural Head Avatars from Monocular RGB Videos". Other relevant 3DGS methods, such as FlashAvatar, which focus on inference acceleration, are also missing.

**Questions:**

1. Why does the method in the article fail to achieve higher FPS? What factors might be limiting the speed?
2. It would be helpful to improve the rendering speed of this method and compare with more baselines.

---

### Official Review · Reviewer_pGG9 · 2024-10-31

**Soundness:** 3
**Presentation:** 2
**Contribution:** 2
**Rating:** 5
**Confidence:** 5

**Summary:**

This study presents a 3D facial avatar model that combines expressiveness with efficiency. Using a compact tensorial feature representation, static color information is stored in tri-planes, while dynamic textures are represented by lightweight 1D feature lines to capture opacity changes. This design achieves accurate, real-time rendering with reduced storage, making it versatile for various applications.

**Strengths:**

1. Separately Designing Dynamic Geometric Deformation and Appearance Change:

This paper builds on GaussianAvatars, which achieves efficient geometric deformation using FLAME’s mesh deformations. However, GaussianAvatars had a notable limitation in representing deformations due to the constraints of coarse 3DMM deformations. This work addresses that limitation by designing dynamic geometric deformations separately from appearance changes.

2. Compact and Efficient Texture Representation:

The use of triplanes and line-tensors improves memory efficiency, offering a more compact and efficient texture representation.

**Weaknesses:**

1. Justification for Dynamic Textures Needed:

Although the authors incorporate dynamic textures, such as blendshape-conditioned opacity, they do not provide a clear justification for their necessity. If I were one of the authors, I would explain the need for dynamic textures as a complementary function to enhance the limitations of coarse 3DMM deformations or to address specific ambient occlusions, such as those for teeth, eyeballs, and the tongue. I recommend that the authors clarify the importance of dynamic textures to reinforce their role in improving model fidelity and handling occlusions effectively.

2. Limitations Not Stated:

No work is without limitations, and it is surprising that this paper does not include a section addressing its constraints. Openly acknowledging limitations is a hallmark of rigorous academic research. As seen in Figure 5, one of the key limitations of this work is identity preservation. For instance, in the first row, the "ours" model exhibits almost no eye specularity, and in the second row, the variation in noise width may give the impression of an identity shift. I suggest that the authors quantify identity similarity by including metrics such as ArcFace or other face recognition scores to verify "no identity shift" claims in Tables 1 and 2.


3. No Improved Performance Observed:

After reviewing the supplementary video, I am not convinced that there is a clear performance improvement compared to GaussianAvatars. I suggest that the authors emphasize and highlight results that showcase significant differences from GaussianAvatars to clarify any substantial performance gains.

**Questions:**

No question. I have posted all things in the weakness section.

---

### Official Review · Reviewer_UjxT · 2024-11-03

**Soundness:** 3
**Presentation:** 3
**Contribution:** 3
**Rating:** 6
**Confidence:** 5

**Summary:**

The paper proposes a high-fidelity and efficient dynamic 3DGS-based approach for modeling dynamic 3D head avatars with expressive and detailed facial expressions.

The proposed approach efficiently compresses the neutral canonical and expression space via a triplane and a series of 1D feature lines, respectively. The above allows for rendering high-fidelity dynamic 3D human heads in realtime.

To allow better generalization to less observed expressions (typically large facial expressions), the authors proposed a cluster-based uniform sampling that randomly selects frames within each cluster of a given expression class. The above addresses inherent dataset biases during training.

**Strengths:**

- The is well-structured and well-written.
- Simple yet effective design that finds a good trade-off between quality,  runtime, and storage.
- Simple yet effective design of dynamic textures via 1D feature lines that is able to represent fine-scale details.
- The proposed method advances the SoTA on high-quality generation of dynamic 3D head avatars with realtime capabilities and efficient storage. So, it’s very practical for VR and teleconferencing applications.

**Weaknesses:**

- No discussion of limitations (see Questions section for detailed concerns).
- Advantages of the approach could be further demonstrated by rendering at higher resolutions, e.g., 2K and 4K resolutions.
- Quantitative results don’t justify the use of cluster-based uniform sampling. As such, a better exposition of results or metrics is required (see Questions section for detailed concerns).
- Strong assumptions about the design of the canonical space can cause further artifacts in more dynamic areas with topological changes, such as the mouth interior.
- No discussion of societal negative impact or ethical considerations (see Details of Ethics Concerns for some suggestions).
- Comparison with other recent methods are missing (see Question section for more details).

**Questions:**

*Method section
- Section 3.1: Most researchers are now familiar with 3DGS preliminaries. As such, I would suggest moving this section to the appendix.
- Fig 2. is confusing. According to the high-level diagram overview, the 3DGS rasterizer should only produce a human head at a neutral pose, while the output with the opacity offset should produce an expressive 3D human head. Or am I missing something? I believe the direction of the arrow pointing toward the opacity offset is wrong (it should be the other way around) based on the method description. Please comment and fix it accordingly.
-  Section 3.2: Lines 243-247. Employing a lower dimension for Txz and Tyz is a strong assumption. While this assumption might work on very structured facial regions, such as the skin surface, it might not work well for more intricate regions that undergo topological changes, such as the mouth interior. Note that the paper does not show novel viewpoint synthesis for extremer 3D head poses either.
Please ablate this assumption (lower vs higher dimension for Txz and Tyz), both quantitatively and qualitatively on frontal and extreme viewpoints (e.g., profile views).

- Class-balanced sampling. Sampling would be more effective if the dataset was split into mild and large expressions. Then, I would run training on a subset with mild expressions until convergence. In a second training run, I would add the large expressions to the dataset and preserve a 20-30% mild and 70-80% large ratio while uniformly sampling frames in each mild and large expression batch.
- Loss function: Please ablate teeth and hair normalization terms quantitatively, i.e., provide a table with metrics in the supplementary material.
I believe the results would be sharper if a gradient loss function is added, i.e., |grad(I)-grad(I’)|, where grad(*) is a Sobel operator, or if a perceptual loss function using low-level feature maps is utilized.

*Experiments section
- Section 4.2: Implementation details:
Please add a high-level architecture design to the supplementary document for triplane + MLP and 1D feature lines + MLP. That would provide a better understanding and reproducibility of the proposed approach.
- Table 2. Results are on par or better w/o the proposed sampling strategy. Based on these results, the sampling is not worth it. Please provide a breakdown of performance metrics and qualitative results for different facial regions and viewpoint ranges to see if this sampling can be justified. Please also include visualizations of the data distribution before and after applying the sampling strategy to illustrate its impact.

*References: Please add the following papers:
- Chen et al. 2023. Implicit Neural Head Synthesis via Controllable Local Deformation Fields. CVPR 2023
- Xiang et al. 2024. FlashAvatar: High-fidelity Head Avatar with Efficient Gaussian Embedding. CVPR 2024
- Saito et al. 2024. Relightable Gaussian Codec Avatars. CVPR 2024
- Teotia et al. 2024. GaussianHeads: End-to-End Learning of Drivable Gaussian Head Avatars from Coarse-to-fine Representations. Siggraph Asia 2024.

Please compare against Saito et al. 204. Optionally compare against Teotia et al. 2024. Both methods adopt 3DGS as the core representation for modeling head avatars, run inference in realtime, and can work with sparse multiview imagery.

*Limitations: Please add a Limitation section and discuss the following potential limitations listed below:
- Can the proposed approach handle torso movements and head rotations?
- The proposed approach doesn’t seem to handle topological changes in the mouth interior well. Please comment and show results zoomed in to the mouth interior for different viewpoints.
- Can the proposed approach handle fast facial expression activations?
- Can this method effectively work at 2K and 4K resolutions? If so, please show results at 2K and 4K resolutions.

**Details Of Ethics Concerns:**

High-quality 3D avatar animation is a hot topic and can be misused. As researchers, we should take societal negative impact seriously. Please add a brief section on ethical considerations. In there, please cover potential misuse scenarios, e.g., deepfakes, and discuss additional mitigation strategies or guidelines for responsible use of the technology and detection of synthetically generated results.

---

### Note · Authors · 2024-11-14

I have read and agree with the venue's withdrawal policy on behalf of myself and my co-authors.